# Critical spin fluctuations across the superconducting dome in $La_{2-x}Sr_xCuO_4$

Jacopo Radaelli [1] ✉, Aavishkar A. Patel [2,3], Mengze Zhu [1], Oliver J. Lipscombe[1], J. Ross Stewart [4], Subir Sachdev [5] & Stephen M. Hayden [1] ✉

Overdoped cuprate superconductors are strange metals above their superconducting transition temperature. In such materials, the electrical resistivity has a strong linear dependence on temperature ($T$) and electrical current is not carried by electron quasiparticles as in conventional metals. Here we demonstrate that the strange metal behaviour co-exists with strongly temperature-dependent critical spin fluctuations showing dynamical scaling across the cuprate phase diagram. Our neutron scattering observations and the strange metal behaviour are consistent with a spin density wave quantum phase transition in a metal with spatial disorder in the tuning parameter. Numerical computations using a theory of spin density waves in a disordered metal yield an extended 'Griffiths phase' with scaling properties in agreement with experimental observations. Thus we establish that low-energy spin excitations and spatial disorder are central to the strange metal behaviour.

Understanding high-temperature superconductivity in layered cuprates has posed significant challenges to the theory of condensed matter. In particular, the unusual metallic state of cuprates above their critical temperature, $T_c$, remains poorly understood. This state is crucial as superconductivity emerges from it. The underdoped region of the cuprate phase diagram is characterized by the presence of various competing orders such as antiferromagnetism (AF), charge density wave (CDW) order and the pseudogap state[1]. In contrast, the overdoped region is free from competing phases but its normal state ($T > T_c$) shows "strange metal" (SM) behaviour[2] where the resistivity, $\rho$, is proportional to temperature, $T$, with a large proportionally constant.

The resistivity of a metal is usually interpreted using the Drude model in which $\rho \propto \tau_{tr}^{-1}$, where $\tau_{tr}^{-1}$ is the relaxation rate of the electrons transporting current. SM behaviour occurs in many classes of material[3] where it has been found that $\tau_{tr}^{-1}$ is approximately equal to $k_B T/\hbar$, the Planckian dissipative rate[4]. This is in stark contrast to the $\rho \propto T^2$ and $\tau_{tr}^{-1} \propto T^2$ expected to describe metals in Fermi liquid theory at low temperature. The SM[3] in resistivity can be accompanied by an

anomalous specific heat contribution $C \propto T \ln(1/T)$ in contrast to the liquid Fermi form $C \propto T$.

The existence of SM behaviour over a wide $T$ range in cuprates is challenging for theory to explain[5]. The picture in simple metals is based on the current-carrying fermion quasiparticles near the Fermi surface scattering off bosons (e.g., phonons). Phonons can only give a linear behaviour at high temperatures $T \gtrsim \theta_{Debye}$ where all phonon modes are excited. Electron-electron scattering yields $\rho \propto T^2$ because only electrons within $k_B T$ of the Fermi surface scatter. In addition, the Planckian dissipation mentioned above implies that quasiparticles lose coherence in the shortest time allowed by quantum mechanics and that the quasiparticle concept breaks down.

Here we show that nearly-critical low-energy collective spin fluctuations are central to the strange metal behaviour in the cuprates and exist across the superconducting dome. We investigate the overdoped region of a cuprate superconductor where the SM behaviour is clearest[6]. The material we have chosen is $La_{2-x}Sr_xCuO_4$ (LSCO). This system can be hole-doped over a wide range through the overdoped region of the phase diagram (Fig. 1A) and has a relatively low $T_c$ so that

[1]H.H. Wills Physics Laboratory, University of Bristol, Bristol, United Kingdom. [2]International Centre for Theoretical Sciences, Tata Institute of Fundamental Research, Bengaluru, India. [3]Center for Computational Quantum Physics, Flatiron Institute, New York, USA. [4]ISIS Pulsed Neutron and Muon Source, Rutherford Appleton Laboratory, Didcot, United Kingdom. [5]Department of Physics, Harvard University, Cambridge, USA. ✉e-mail: jacopo.radaelli@bristol.ac.uk; s.hayden@bristol.ac.uk

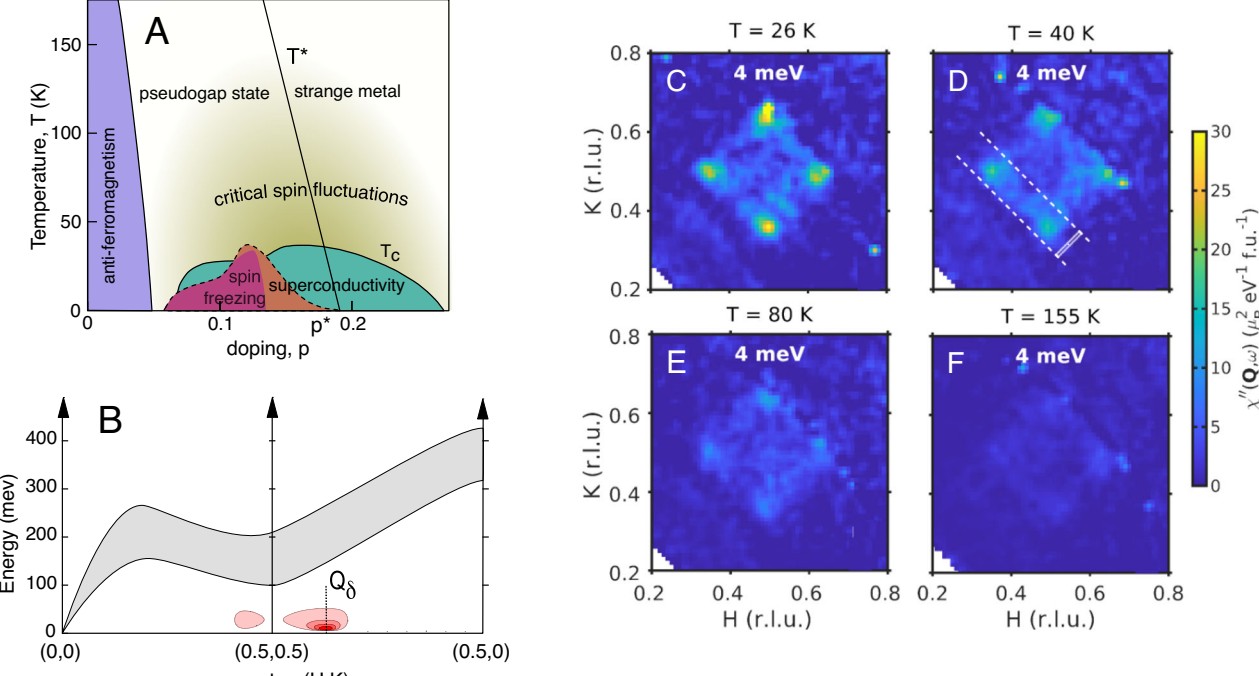

**Fig. 1 | Spin fluctuations in La$_{2-x}$Sr$_x$CuO$_4$ (x = 0.22). A** Schematic phase diagram of LSCO showing the possible extent of the critical spin fluctuations observed here. The dashed line shows the extent of spin freezing for $B$ = 80 T[13] where super-conductivity is fully suppressed and the magenta region for $B$ = 0 from ref. [49]. **B** Schematic of spin excitations in LSCO($x$ = 0.22) based on (refs. [9,10]) as represented by contours of the imaginary part of the susceptibility $\chi''(\mathbf{Q}, \omega)$. Reciprocal space is labelled as $\mathbf{Q} = H\mathbf{a}^\star + K\mathbf{b}^\star + L\mathbf{c}^\star \equiv (H, K)$. The gray region shows the broad high-energy spin excitations[9,10]. Pink regions are the low-energy excitations studied here. **C–F** Slices of $\chi''(\mathbf{Q}, \omega)$ for $\hbar\omega$ = 4 meV, $L \in [-1, 1]$ and various temperatures (see Methods for details). The white rectangle in (D) shows the region of integration used to produce the points in 1-D cuts such as those in Fig. 2A, with integration along (1,1). Dashed lines represent the path of the 1-D cuts.

the SM normal state is present over approximately two orders of magnitude in temperature.

## Results

### Spin fluctuations in cuprates

Spin fluctuations have been widely studied in LSCO and other cuprate superconductors by techniques such as inelastic neutron scattering, resonant inelastic x-ray scattering and nuclear magnetic resonance[7]. The parent compounds such as La$_2$CuO$_4$ are antiferromagnetic with strong super-exchange coupling, $J$, and spin wave excitations[8] up to ~320 meV. Upon doping these spin wave-excitations become heavily damped, with residual antiferromagnetic or spinon excitations remaining at higher energies $\gtrsim$100 meV[9,10] as illustrated schematically in Fig. 1B. In addition to the band of higher energy excitations, doped cuprates show low-energy excitations which are highly structured in reciprocal space[11] with the strongest excitations at $\mathbf{Q}_\delta = (1/2, 1/2 \pm \delta)$ and $(1/2 \pm \delta, 1/2)$. For underdoped compositions, with doping $x = p \sim 1/8$, these incommensurate spin fluctuations freeze for $T \lesssim T_c$[12–14]. At higher temperatures in the normal state they are strongly temperature dependent showing the hallmarks of proximity to quantum criticality such as $\omega/T$-scaling[15–17]. For overdoped LSCO, low-energy spin fluctuations were observed to grow monotonically down to $T \approx T_c$[18]. For LSCO($x$ = 0.22) and $T \approx T_c$ = 26 K, it was recently found[19] that these excitations are described by a heavily over-damped harmonic oscillator response and have a characteristic energy scale $\hbar\Gamma_\delta \approx 5$ meV $\approx 3k_BT$.

We used inelastic neutron scattering to map out the $\mathbf{Q}$-$\omega$ dependence of the low-energy spin fluctuations in single crystals of LSCO($x$ = 0.22) in the normal state for temperatures $T_c \le T \le 300$ K. Measurements were carried out using the LET spectrometer of the ISIS Pulsed Neutron and Muon Source (See Methods). Figure 1C–F show constant energy maps of the dynamical susceptibility $\chi''(\mathbf{Q}, \omega)$ at $\hbar\omega$ = 4 meV. For $T$ = 26 K, the four-peaks around $\mathbf{Q}$ = (1/2, 1/2) can

be clearly seen in Fig. 1C. Cuts through the peaks along the $\mathbf{Q}$ = (1/2 − $\delta$/2 + $\xi$, 1/2 − $\delta$/2 + $\xi$) line (along the centre of dashed lines in Fig. 1D) are shown in Fig. 2A. As the temperature increases, the peaks can be seen to weaken and broaden in $\mathbf{Q}$. The measured intensity $S(\mathbf{Q}, \omega)$ can be converted to the the magnetic response function $\chi''(\mathbf{Q}, \omega)$ using the fluctuation dissipation theorem $S(\mathbf{Q}, \omega) = (1/\pi)\chi''(\mathbf{Q}, \omega)[1 - \exp(-\hbar\omega/k_BT)]^{-1}$ (See Methods). This is shown in Fig. 2B and Fig. 3A. Our data (see Fig. 3A) show that the dynamical spin susceptibility near $\mathbf{Q}_\delta$ is strongly temperature dependent for this overdoped cuprate. The behaviour is reminiscent of the approach to a quantum critical point as $T \to 0$.

### Modelling the low-energy spin fluctuations

At a quantum critical point, the dynamical susceptibility is expected to have the following scaling form[20]

$$\chi(q, \omega) \propto T^{-\alpha} \Phi\left(\xi q, \frac{\hbar\omega}{k_BT}\right), \tag{1}$$

where $\xi \propto T^{-\frac{1}{z}}$ is a correlation length, $z$ is the dynamical critical exponent, $\mathbf{q} = \mathbf{Q} - \mathbf{Q}_\delta$ is the wavevector relative to the ordering wavevector, $q = |\mathbf{q}|$ and $\Phi$ is a universal complex function of both arguments. For $q = 0$, the imaginary part of the dynamical susceptibility (measured here) has the following scaling form (See Supplementary Methods):

$$\chi''(q = 0, \omega) = T^{-\alpha}\phi_1\left(\frac{\hbar\omega}{k_BT}\right), \tag{2}$$

where $\alpha = \gamma/\nu z$, $\phi_1(x) = \Im[\Phi(0, x)]$ is a scaling function and $\gamma$ and $\nu$ are the susceptibility and correlation function temperature exponents respectively. This is an example of $\omega/T$-scaling.

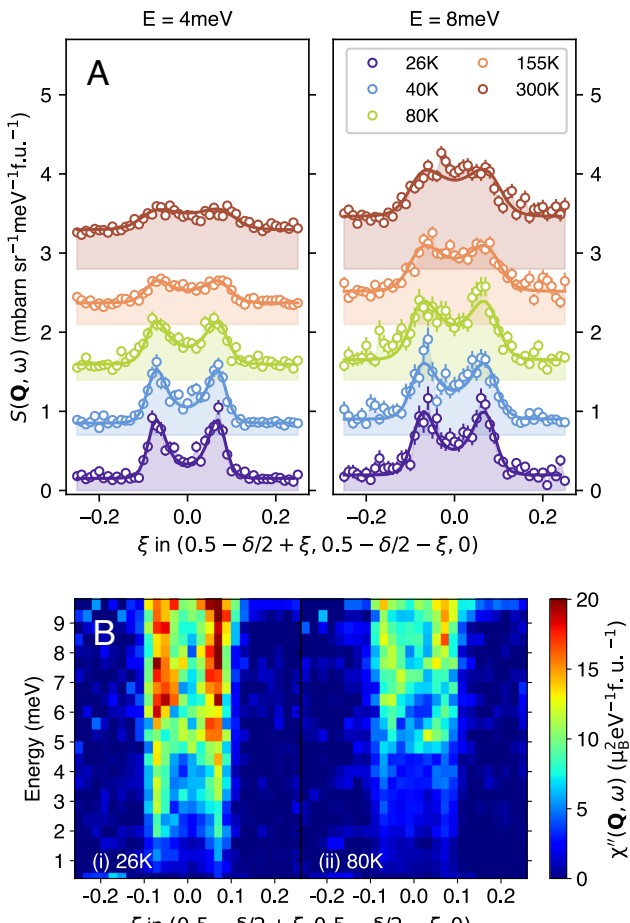

**Fig. 2 | The temperature dependence of the low-energy spin fluctuations in La$_{2-x}$Sr$_x$CuO$_4$ (x = 0.22). A** The scattering intensity versus wavevector **Q** for different temperatures (bottom to top: 26, 40, 80, 155, 300 K) for energy transfers $\hbar\omega = 4, 8$ meV. The cuts are through the incommensurate **Q**$_\delta$ positions as shown by the dashed lines in Fig. 1D. **B** $E - \mathbf{Q}$ map of the magnetic response function $\chi''(\mathbf{Q}, \omega)$ for $T = 26, 80$ K showing its evolution with temperature. The trajectory of **Q** is the same as that in (**A**). Error bars are determined from Poisson statistics and the neutron counts. They are one standard deviation.

In Fig. 3B we show that our data can be collapsed onto a single trend with a suitable choice of $\alpha$. An analogous collapse is observed in heavy fermion systems[21,22]. In our case, $\phi_1(x)$ can be approximated by a simple Lorentzian $\phi_1(x) \propto ax/(a^2 + x^2)$ with $a = 2.9 \pm 0.3$ and $\alpha = 0.32 \pm 0.10$. This small value of $\alpha$ excludes the familiar paramagnon theory which has $\alpha = 1$[23,24] and does not show an extended regime of criticality. Our data are consistent with the measured susceptibility having a low-frequency component that varies as

$$\chi''(\mathbf{Q}_\delta, \omega) = \frac{\chi'(\mathbf{Q}_\delta)\Gamma_\delta\,\omega}{\Gamma_\delta^2 + \omega^2}, \qquad (3)$$

where the real part of the susceptibility $\chi'(\mathbf{Q}_\delta) \propto T^{-\alpha}$ and the spin relaxation rate varies as

$$\hbar\Gamma_\delta = ak_BT, \qquad (4)$$

as shown in Fig. 3A (inset). Thus our data are consistent with $\Gamma_\delta$ tracking the Planckian relaxation rate $\tau_{tr}^{-1} \propto T$ as observed in transport measurements suggesting that they are related.

More information about the criticality of the spin fluctuations can be gained from the **Q**-dependence of $\chi''(\mathbf{Q}, \omega)$. We fitted our data using a form previously used[17,19] to describe the low-energy response of the

cuprates:

$$\chi''(\mathbf{Q}, \omega) = \chi''(\mathbf{Q}_\delta, \omega) \frac{1}{[1 + \kappa^{-2}(\omega)R(\mathbf{Q})]^2}, \qquad (5)$$

where $R(\mathbf{Q})$ is a function with zeros at the $\mathbf{Q}_\delta$ positions and $R(\mathbf{Q}) = |\mathbf{q}|^2$ near $\mathbf{Q} = \mathbf{Q}_\delta$ [see Supplementary Equation (11)]. Eqn. (5) has peaks at the four $\mathbf{Q}_\delta$ positions and $\kappa(\omega)$ is a measure of the peak width. Example fits to Eqn. (5) are shown in Fig. 2 and the resulting values of $\kappa(\omega)$ are shown in Fig. 3D–F. Eqns. (1) and (5) taken together (See Supplementary Methods) imply an additional $\kappa$-scaling

$$\kappa(\omega) = T^{\frac{1}{z}}\phi_2\left(\frac{\hbar\omega}{k_BT}\right) \qquad (6)$$

and $\kappa(0) \propto T^{\frac{1}{z}}$.

In Fig. 3D we see that $\kappa(\omega)$ increases with $\omega$ and $T$. We posit that the effects of $\omega$ and $T$ add as

$$\kappa^2(\omega) \propto (\hbar\omega)^{2/z} + r(k_BT)^{2/z}, \qquad (7)$$

where $r$ is the ratio of the temperature and energy contributions which we set equal to one. This form satisfies Eqn. (6). We find that Eqn. (7) can be used to collapse the experimentally determined $\kappa(\omega, T)$ onto a single solid line as shown in Fig. 3E, F with $z = 1.83 \pm 0.35$. In the Supplementary Methods we show that our data are also consistent with the system being tuned slightly away from quantum criticality (dashed line in Fig. 3F) within the uncertainty of the experiment.

Here we only study the normal state, however, we can extrapolate the data to lower temperatures. Figure 3 suggests that the spin fluctuations would continue to evolve as $T \to 0$ in the absence of superconductivity. This is supported by measurements made where superconductivity is suppressed by a magnetic field. Previous neutron scattering measurements[19] show that $\Gamma_\delta$ continues to decrease below $T_c$ when a 9 Tesla field is applied. High field ($B \gtrsim B_{c2}$) heat capacity measurements show a $\sim T\log(1/T)$ contribution[25] and resistivity measurements in a high field[6] show a $\rho \propto T$ behaviour for $T$ going to zero.

Thus we have found that the low-energy spin fluctuations of the overdoped superconductor LSCO($x = 0.22$) show the hallmarks of quantum criticality. Specifically, we observe that both $\chi''(q = 0, \omega)$ and $\kappa(\omega)$ show $\omega/T$-scaling. The dynamical critical exponent inferred is $z = 1.83 \pm 0.35$. Our value of $z$ for LSCO($x = 0.22$) is consistent with the $z = 2$ value expected[23,24] for a 2D antiferromagnetic metal at a QCP with coupling between electrons and spin fluctuations. It is important to view this result in the context of the whole cuprate phase diagram. Previous $T$-dependent measurements in the normal state of underdoped LSCO($x = 0.14$)[17] show a related $\omega/T$-scaling when replotted and analysed in same way as the present data (see Fig. 3C). The LSCO($x = 0.14$) data has a similar value of $\alpha \approx 0.32$ and a smaller dynamical critical exponent, $z \approx 1$, suggesting that $z$ increases with doping. Combining the results of the two studies we conclude that the normal state hosts low-energy critical fluctuations across the phase diagram as illustrated in Fig. 1A.

## Discussion

Understanding the strange metal behavior seen in LSCO and other cuprates is one of the major challenges in condensed matter physics. It occurs across a diverse set of materials that include transition metal oxides, heavy fermions[21,26], iron-based superconductors and twisted bilayer graphene[2,3]. Planckian dissipation provides a universal phenomenology representing a quantum limit on the current-carrying degrees of freedom of a metal[4]. However, it does not indicate a microscopic mechanism. The behaviour in the resistivity has often been associated with an underlying magnetic quantum critical point

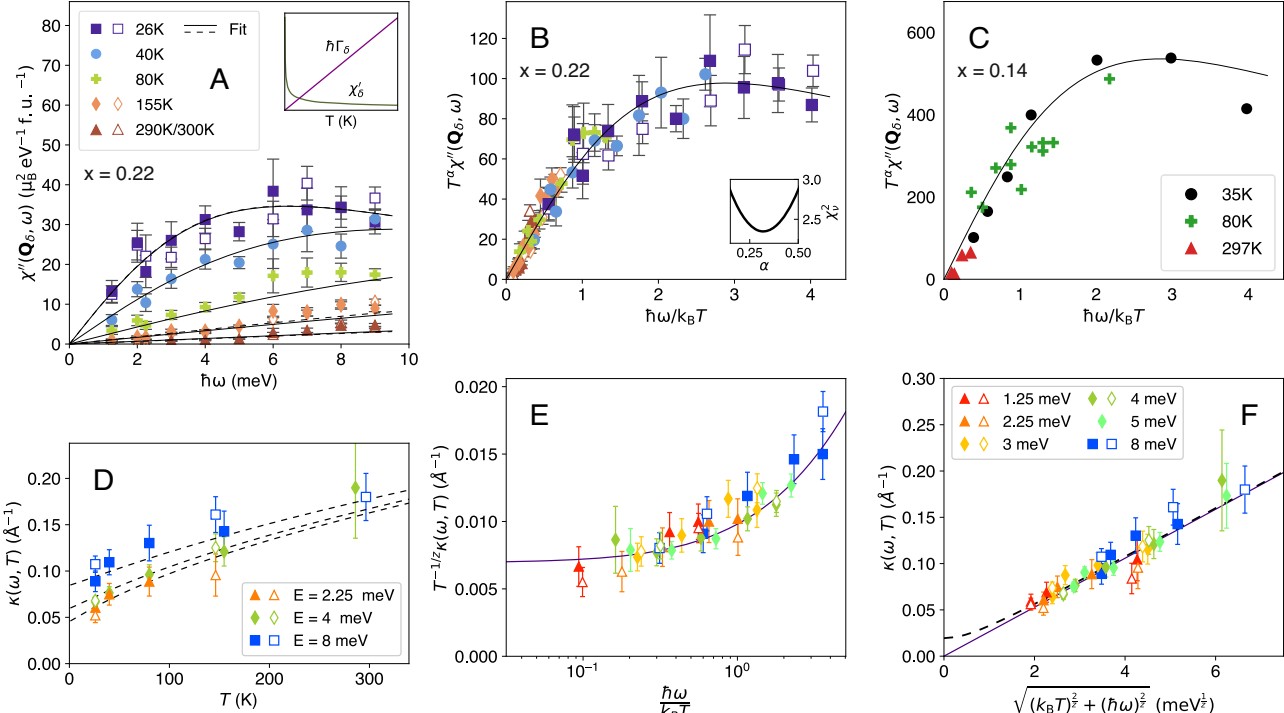

**Fig. 3 | Scaling behaviour of the dynamical magnetic susceptibility and correlation length in La$_{2-x}$Sr$_x$CuO$_4$. A** Temperature dependence of $\chi''(\mathbf{Q}, \omega)$ at ordering wavevector $\mathbf{Q}_\delta$. Lines are fits to Eqn. (3). Open and closed symbols are two separate experiments. Inset shows the $T$-dependence of the spin relaxation rate $\Gamma_\delta$ and real part of the susceptibility $\chi'(\mathbf{Q}_\delta)$ obtained when data are modelled with Eqns. (3)–(4). **B** $\omega/T$-scaling plot of the $x = 0.22$ data in (A) using Eqn. (2) and the procedure described in Methods. The inset shows how the quality of the collapse varies with exponent $\alpha$. **C** Scaling plot for underdoped LSCO $x = 0.14$ data from ref. 17. This data shows a similar collapse to the $x = 0.22$ data in (**B**). **D** The width in **Q** of $\chi''(\mathbf{Q}, \omega)$ is denoted by the inverse dynamical correlation length $\kappa(\omega)$. Here we plot $\kappa(\omega)$ versus temperature for $\hbar\omega = 1.25, 2.25, 4, 8$ meV. **E** Data in (**D**) can be collapsed onto a single trend line with a suitable choice of $z$ and $\omega/T$-scaling. **F** Data in (**D**) can be scaled onto a single line using (7) allowing the dynamical critical exponent $z$ to be determined. Dashed line is Supplementary Equation 16. Error bars are determined statistically from fitting and are one standard deviation.

(QCP) at $T = 0$. In a typical scenario the system is tuned to the QCP and strange metal behaviour is observed over a relatively small range of parameter space. This is observed in some systems such as Sr$_3$Ru$_2$O$_7$[27,28]. However, there are many examples including not only cuprates but twisted-layer graphene and twisted transition-metal dichalcogenides where it persists over of an extended region. Thus alternative explanations must be explored.

The phase diagram of LSCO consists of a region of glassy incommensurate spin-freezing[12–14,29] near $p \approx 1/8$ (also seen in other cuprates[30]). The spin freezing competes with superconductivity and when the superconductivity is suppressed by a large magnetic field $B \sim B_{c2}$ the region of spin freezing is expanded to $p \approx 0.19 \approx p^\star$ (see Fig. 1A). This region reflects how the system might behave in zero field if the superconducting state did not form. The presence of an extended region of criticality as a function of doping at higher temperatures together with a disappearing frozen state at low temperatures is reminiscent of a 'quantum Griffiths phase'[31–33] as illustrated in Fig. 4A. A Griffiths phase[31,33] occurs near a continuous phase transition in systems with quenched disorder. The inclusion of disorder is logical in the La$_{2-x}$Sr$_x$CuO$_4$ system because of the perturbation caused by Sr doping in the plane neighbouring the CuO$_2$ plane. The Griffiths phase[31,33] corresponds to a smearing of singular behaviour as a function of a control parameter. It can naturally lead to an extended region of slow dynamics as observed in LSCO.

Here we consider the scenario where the disorder in LSCO leads to Griffiths behaviour and can be modelled by a Hertz-Millis theory[24,34] for the onset of magnetic order in metals in the presence of spatial disorder in the tuning parameter. The dominant source of disorder in such a theory is of the 'random mass' type, by which local regions are tuned away from the quantum critical point of the clean system. As

reviewed in the Supplementary Methods, such a theory was studied by a numerically exact treatment of disorder but with a mean-field treatment of self-interactions of the magnetic order fluctuations[35]. This treatment is inspired and justified by the mapping of such models onto two-dimensional Yukawa-Sachdev-Ye-Kitaev models (2D-YSYK), and the latter have been treated by SYK methods and exact quantum Monte Carlo simulations, with similar results[5,36–38].

It has been shown that the inclusion of disorder (in the 2D-YSYK and related models) can explain the existence of the SM-behaviour down to the lowest temperatures and over an extended range of doping[35,38]. Applying this general picture to the cuprates, it is postulated that disorder delays the onset of long-range spin order giving rise to a "quantum Griffiths phase"[32,35,38–40] (similar to that in the two-dimensional Ising model in a transverse field[41]) as a transition tuning parameter $\lambda$ is reduced. SM linear-in-$T$ resistivity is observed over an extended range of $\lambda$ that comprises part of the Griffiths phase instead of a QCP. The tuning parameter $\lambda$ is controlled by the doping $x$.

In addition to the strange metal behaviour in the resistivity observed in LSCO and other cuprates, another important property is the optical conductivity $\sigma(\omega)$. An extended Drude analysis shows that the optical scattering rate $\tau_{op}^{-1}(\omega)$ is strongly frequency dependent[42,43] indicating the current-carrying electrons are scattered inelastically. This is incompatible with elastic scattering from impurities or defects. Phonon scattering cannot produce the $T$-linear resistivity at low temperature below the Debye temperature scales. This leads to the conclusion that disorder in strange metals must scatter electrons inelastically, which is predicted by the 2D-YSYK theory[5,37]. The optical scattering rate $\tau_{op}^{-1}(\omega)$ provides a further insight into the nature of strange metals as it is found to show $\omega/T$-scaling (along with the magnetic response reported here) over a wide range of $\omega/T$[43].

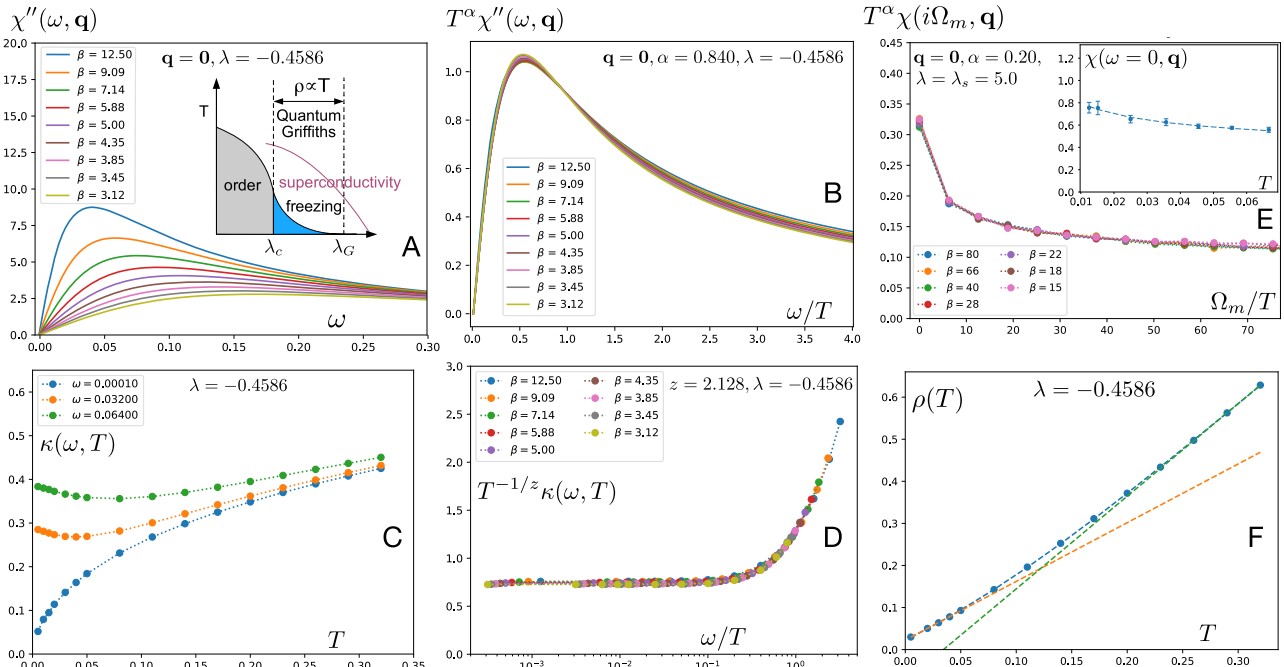

**Fig. 4 | Numerical theoretical results for low energy spin fluctuations.**
**A**, **B** Dynamical spin susceptibility and its scaling plot with best-fit exponent $\alpha = 0.84$. The inset in panel **A** provides a schematic phase diagram of the model.
**C**, **D** Inverse correlation length $\kappa$ and its scaling plot with best-fit critical exponent $z = 2.128$. $\kappa$ is in units of the inverse lattice spacing. We estimate $\approx 33$ meV as the unit of energy/frequency/temperature in the numerics and $\beta = 1/T$. Theory plots should be compared with the experimental plots in Fig. 3A–E. Plots based on a Hertz-Millis model with spatial disorder are obtained from the results of ref. 35 (with a mean-field treatment of interactions, but exact treatment of disorder), at the critical value of $\lambda = \lambda_c = -0.4586$, after exact analytic continuation to real frequencies (See Supplementary Methods); results for $\lambda > \lambda_c$ are in the Supplementary Methods.
**E** Computations of imaginary frequency ($\Omega_m$) susceptibility by (in principle, exact)

quantum Monte Carlo (QMC) results for the YSYK model in ref. 38 at the end of the strange metal quantum Griffiths phase that is present for $\lambda_s = 5.0 \leq \lambda \leq \lambda_G = 5.5$. In the QMC results, the transition into an ordered phase at the critical point $\lambda_c$ is replaced with a transition into a phase with glassy short-range order at $\lambda_s$. The dashed line in the inset is a fit to $- T^{-\alpha}$. Both plots yield $\alpha = 0.20$. The energy unit for the QMC computations is the fermion hopping $t \approx 0.3$ eV. **F** Resistivity from spin fluctuations. The resistivity is computed from the spin susceptibility as described in ref. 35. Note the distinct slopes of linear-in-$T$ resistivity at low and high $T$. Using the energy unit of $\approx 33$ meV and the parameters of ref. 37, with the fermion hopping $t \approx 0.3$ eV and the variance of the random Yukawa coupling $g'^2 = 4t$, the largest resistivity shown in the plot is $\approx 0.15\, h/e^2$.

In Fig. 4A–D we show numerical calculations of the critical spin susceptibility $\chi''(\mathbf{Q}_\delta, \omega)$ from the Hertz-Millis model with spatial disorder in the tuning parameter. The form in Eqn. (5) is an excellent fit to this theory (See Supplementary Methods) and allows the determination of $\kappa(\omega, T)$, which is also shown. The results have been scaled in the same way as Fig. 3B, E and we find excellent real frequency scaling at the quantum critical value of the tuning parameter $\lambda = \lambda_c = -0.4586$ with $\alpha = 0.84$ and $z = 2.13$. Corresponding scaling plots within the quantum Griffiths phase, $\lambda_G \geq \lambda > \lambda_c$, appear in the Supplementary Figs. 1–8: the $\omega/T$-scaling is still present, but it is better at smaller values of $\omega/T$, and for the local spin susceptibility. The better scaling of the local susceptibility is as expected from the localized nature of the spin fluctuations in the Griffiths region. The value of the exponent $\alpha$ decreases significantly with increasing $\lambda$, down to $\alpha \approx 0.5$ at the beginning of the quantum Griffiths phase where $\lambda = \lambda_G = -0.43$. A separate, and in principle, exact determination of $\alpha$ is obtained from the imaginary frequency ($\Omega_m$) susceptibility of the quantum Monte Carlo results of ref. 38 (Fig. 4E). This also obeys $\Omega_m/T$ scaling and yields $\alpha \approx 0.2$ in the Griffiths region, compatible with our observations at $x = 0.22$. We note that $\omega/T$-scaling fails in our theory at very low $T$ (See Supplementary Methods), as it does in the Ising model in a transverse field[41], presumably due to the dominance of rare regions: future observations at even lower $T$ are therefore of interest.

We now turn to the resistivity $\rho(T)$ induced by the disordered spin fluctuations. In 2D-YSYK theory, $\rho(T)$ is determined by the imaginary part of the fermion self energy from spin fluctuations[35]. By computing this self energy, the Hertz-Millis model with spatial disorder in the tuning parameter is able to produce a linear resistivity and Planckian

behaviour in the transport and optical scattering rates[35] yielding a consistent picture of the strange metal, as shown in Fig. 4F, with similar results at other values of $\lambda$ in the quantum Griffiths phase shown in the Supplementary Methods. There is linear-in-$T$ behaviour, but with different slopes at lower and higher $T$: we can identify these with the 'foot' and 'fan' regimes, associated with Griffiths and marginal Fermi liquid behaviours respectively[6,43,44]. The low $T$ upturn in Fig. 4C in $\kappa$ for $\omega > T$ is possibly also related to the crossover between these 'foot' and 'fan' regimes.

In summary, we have observed critical spin fluctuations with $\omega/T$-scaling in an overdoped cuprate superconductor. When combined with earlier measurements[17], this shows that there is an extended region of critical spin-fluctuations in La$_{2-x}$Sr$_x$CuO$_4$. We have compared our observations with numerical studies of a Hertz-Millis model for the onset of spin density wave order in a disordered metal and found good agreement. Both experiment and theory/numerics find an extended Griffiths phase of both $\omega/T$ magnetic susceptibility scaling [in $\chi''(\mathbf{Q}_\delta, \omega)$ and $\kappa(\omega)$] and $T$-linear resistivity. Such scaling is present even though there is no criticality in the usual sense of a diverging correlation length (See Supplementary Methods), and is accompanied by scaling deviations consistent with Griffiths physics, where scaling is violated by a logarithmic dependence on energy scales or doping[32,38,40]. The correlation length diverges at the lower doping boundary of the Griffiths region, and the large correlation length allows definition of a finite dynamical exponent $z$ which is doping (tuning parameter) dependent.

The spin excitations show $z \approx 2$, and have a contribution consistent with a spin relaxation rate $\hbar\Gamma_\delta \approx 3k_BT$. The observations may be related to singular charge density fluctuations observed in EELS[45]. The

strongly $T$-dependent spin fluctuations coexist with transport properties described by Planckian dissipation, showing the spin degrees of freedom in the presence of spatial disorder are at the heart of strange metal behaviour. It would be interesting to study the role of such critical spin fluctuations in high temperature $d$-wave superconductivity[46]: it is plausible that the localization of the spin fluctuations plays a role in the near co-incident disappearance of strange metal behaviour and superconductivity with increasing doping in the cuprates[47].

## Methods

### Single-crystal sample growth and characterization
Single crystals of La$_{2-x}$Sr$_x$CuO$_x$ (x = 0.22) were grown by the travelling-solvent floating-zone method. The crystals were annealed in 1 bar of flowing oxygen at 800 °C for six weeks. The Sr concentration was determined by scanning electron microscopy with electron probe microanalyzer (SEM-EPMA) and inductively coupled plasma atomic emission spectroscopy (ICP-AES) to be x = 0.215 ± 0.005. SQUID magnetometry measurements show that $T_{c,onset}$ = 26K. The sample consists of 29.8 g of LSCO crystal, mounted and co-aligned using the ALF single crystal diffractometer.

### Inelastic neutron scattering
Inelastic neutron scattering measurements were performed at the LET direct geometry time-of-flight spectrometer at ISIS. LET is a multiplexing instrument which allows for simultaneous collection of data at multiple neutron incident energies. Three fixed incident energies $E_i$ = 3.76, 6.82, 16.02 meV were used for the whole data collection.

To this end, the sample was mounted with its $c$-axis vertical and the azimuthal angle swept over a range of ~ 108° about the region of interest. Measurements were taken at one degree intervals to ensure adequate coverage over the entire region around the (1/2,1/2) wavevector.

Two experiments were performed. In 'Experiment 1' we measure at 5 different temperatures including room temperature (290K) and the superconducting transition temperature (26K). In Experiment we measured at 3 of the 5 original temperatures with the sample rotated by 90° with respect to Experiment 1 and counted for a longer period at 300 K.

### Data Analysis
The scattering cross-section is related to the scattering function $S(\mathbf{Q}, \omega)$ and energy- and wavevector-dependent magnetic response function $\chi''(\mathbf{Q}, \omega)$ by the fluctuation-dissipation theorem

$$\begin{aligned}\frac{k_i}{k_f}\frac{d^2\sigma}{d\Omega\,dE} &= S(\mathbf{Q}, \omega) \\ &= \frac{2(\gamma r_e)^2}{\pi g^2\mu_B^2}|F(\mathbf{Q})|^2\frac{\chi''(\mathbf{Q},\omega)}{1-\exp(-\hbar\omega/k_BT)} + B.G.,\end{aligned} \quad (8)$$

where $(\gamma r_e)^2$ = 0.2905 barn sr$^{-1}$ and $F(\mathbf{Q})$ the magnetic form factor. B.G. is non-magnetic background scattering such as incoherent inelastic phonon scattering and multiple scattering. Counts measured at position sensitive detectors were normalized to a vanadium standard to correct for differences in detector efficiency and then used to reconstruct the momentum and energy-dependent scattering function $S(\mathbf{Q}, \omega)$. This process was repeated for each $E_i$ producing three 4-D datasets at each temperature.

**1-D Q Cuts and fitting.** In order to parameterise the excitations for a given $\omega$ and $T$, $S(\mathbf{Q}, \omega)$ data slices are binned into 1-D $\mathbf{Q}$-dependent cuts. Cuts were made primarily through the low $Q$ peaks at (1/2,1/2 − δ) and (1/2 − δ,1/2), with similar results obtained from cuts through (1/2,1/2 + δ) and (1/2 + δ,1/2). The cuts are generated by integrating along the $\mathbf{Q}$-direction perpendicular to the cut (see Fig. 1D) in the $H − K$ plane. The integral along $\mathbf{c}^\star$ was $\Delta L$ = ± 1 and in energy $\Delta E$ = ± 0.5 meV. The

trajectory of the cut minimises the variation of $Q$ and hence the variation of background due to incoherent one-phonon scattering.

We carry out a resolution-corrected least-squares fits (see Fig. 2A and Supplementary Fig. 10) to each cut using the Tobyfit module in Horace[48]. The susceptibility is modelled by Eqn. (5) and Eqn. (8). There are three energy-dependent parameters: $\chi''(\mathbf{Q}_\delta, \omega)$ and $\kappa(\omega)$ which control the height and width of the peaks in $\mathbf{Q}$ respectively, and a $\mathbf{Q}$-independent (constant) background. From the fits in Fig. 2A and Supplementary Fig. 10 it can be seen that the background increases with $\omega$ and $T$ consistent with one-phonon incoherent scattering from the sample.

The incommensurability $\delta$ of the excitations also enters into the model via $R(\mathbf{Q})$. The value of $\delta$ in Eqn. (5) and Supplementary Equation (11) was fixed for each temperature. The fitted values ranged from $\delta$ = 0.139 to $\delta$ = 0.131 for $T$ = 26 and 300 K respectively. When making $T$-dependent (scaling) plots we evaluated the fitted $\chi''(\mathbf{Q}_\delta, \omega)$ for the $T$ = 26 K value of $\delta$. This was determined by averaging over energy dependent best-fit values below 5 meV where the peaks are sharp.

### ω/T scaling collapse
For a given $\alpha$ value, we calculated a set of scaled peak susceptibilities $y_i(\omega/T) = T^\alpha\chi_i''(\mathbf{Q}_\delta, \omega/T)$. These were placed in evenly spaced bins in $x = \hbar\omega/k_BT$ between $x = 0$ and $x = 4$. The $y_i$ values were compared against a function $E(x_i)$, calculated using the bin averages and bin centres for that $\alpha$, to give a reduced chi-squared,

$$\chi_v^2 = \frac{1}{N}\sum_{i=1,N}\frac{[y_i - E(x_i)]^2}{\sigma_i^2},$$

where $E(x)$ was a piecewise function with linear segments connecting the $(\tilde{x}_j, \tilde{y}_j)$ points where $\tilde{x}_j$ and $\tilde{y}_j$ are the bin centres and bin averages respectively. We then determine the $\alpha$ that minimises $\chi_v^2$.

**2-D Q and Q-E Slices.** To produce the 2-D $\mathbf{Q}$ slices for display in Fig. 1C−F, data is integrated over $L = \pm 1$ and $\Delta\hbar\omega = \pm 0.5$ meV. $S(\mathbf{Q}, \omega)$ is converted to $\chi''(\mathbf{Q}, \omega)$ using Eqn. (8) following subtraction of a $\mathbf{Q}$-independent background determined from the 1-D fitting described above. A similar procedure is used for the $\mathbf{Q} − \omega$ plot in Fig. 2C except that $\Delta\hbar\omega = \pm 0.25$ meV and a different background (determined from the 1-D $\mathbf{Q}$-cuts) was used for each energy.

### Phonon Scattering
Inelastic neutron scattering also detects phonon scattering due to the interaction of the neutron with the nucleus. One-phonon coherent scattering yields well-defined peaks in $(\mathbf{Q}, \omega)$ and one-phonon incoherent yields smooth variation in $(\mathbf{Q}, \omega)$. The intensity of both types of one-phonon scattering are proportional to $Q^2$ and the Bose factor $[1 - \exp(-\hbar\omega/k_BT)]^{-1}$.

In order to minimize the effect of phonons, we (i) performed the experiment in the first Brillouin zone, (ii) measured for $\hbar\omega \leq 10$ meV where there are relatively few phonons. The scattering we observe is two dimensional. We used a small integral in $L$ to minimize the phonon scattering. Supplementary Fig. 12 shows the magnetic scattering measured near $L = 0$ compared with the phonons measured near $L = 4$ for a 1-D cut with the same $(H, K)$ values. It can be seen that the phonons and magnetic scattering have very different structures in $(\mathbf{Q}, \omega)$. The magnetic scattering presents as columns of scattering and the phonons as a dispersive mode. The model we use to fit data for a particular $\omega$ [Eqn. (5), Eqn. (8) and Supplementary Equation (11)] has a different shape in $\mathbf{Q}$ to the phonon scattering. It therefore picks out the magnetic scattering as shown Fig. 2A and Supplementary Fig. 10B. However, there are some deviations which may be due to phonon scattering e.g., near $\xi = 0$ for $T = 300$ K and $\hbar\omega = 7$ meV.

## Data availability

Source data for experimental figures are provided at https://doi.org/10.5281/zenodo.18682259. Raw data are at https://doi.org/10.5286/ISIS.E.RB2220248-1 and https://doi.org/10.5286/ISIS.E.RB2410260-1.

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

## Acknowledgements

S.M.H. is grateful to Jörg Schmalian for sharing insights about scaling theory. A.A.P. and S.S. thank Peter Lunts for related collaborations. Neutron beamtime was provided by the ISIS neutron and muon source through proposals RB2220248 and RB2410260. Work was supported by the U.K. EPSRC through grant EP/R011141/1. The Flatiron Institute is a division of the Simons Foundation. S.S. was supported by the U.S. National Science Foundation grant No. DMR-2245246.

## Author contributions

Single crystals were grown and characterised by O.J.L and S.M.H. Neutron scattering measurements were performed by J.R., M.Z., J.R.S., and S.M.H. Data analysis performed by J.R. and S.M.H. Numerical theory performed by A.A.P. and S.S. The paper was written by J.R., A.A.P., S.S., and S.M.H.

## Competing interests

The authors have no competing interests.
