## [Transparent Peer Review file · Nature Communications]

Critical spin fluctuations across the superconducting dome in $\text{La}_{2-x}\text{Sr}_x\text{CuO}_4$

Corresponding Author: Professor Stephen Hayden

Version 0:

Reviewer comments:

Reviewer #1

(Remarks to the Author)

This manuscript describes time-of-flight inelastic neutron scattering on an overdoped single crystal of LSCO $x=0.22$ and its analysis in terms of ω/T scaling and comparison to numerical calculations Yukawa-Sachdev-Ye-Kitaev model. The authors argue that this gives a picture consistent with a quantum phase transition generated by low energy, critical spin fluctuations in the presence of disorder. The argument is also made that a similar picture will generate Planckian transport with resistivity $\sim T$ over an extended regime of the overdoped 214 phase diagram – as have been observed.

I found the experiment portion of the manuscript relatively straightforward to follow and well written. I also felt that some of what I would view as important experimental detail are relegated to the supplemental information or otherwise not easily accessible.

I think that the experimental quality of the data is high, and the paper is addressing a topical and interesting subject in contemporary quantum materials physics. Provided the following points can be addressed appropriately, I feel that a suitably modified version of the manuscript is likely appropriate for Nature Communications.

It is clear that with the isolation of $X''(Q, \omega)$ shown in Fig. 1 C-F, Fig. 2 B and Fig. 3 A a very satisfactory ω/T scaling can be achieved with this data set, as shown in Fig. 3 B for LSCO $x=0.22$. A similar analysis is conveniently reproduced here in Fig. 3 C for LSCO $x=0.14$ using earlier work. From that point on the comparison to relevant theory and the idea of a broad Quantum Griffiths Phase is described and developed.

My primary concern/criticism with this manuscript is the first step of isolating $X''(Q, \omega)$ from the measured inelastic intensity. Everything else in the manuscript depends critically on this first step being done correctly so that the magnetic scattering alone is isolated (from background, phonons) correctly. So I would like more detail included on this. This main concern really boils down to how confident are the authors that they have isolated the magnetic scattering alone in their determination of $X''(Q, \omega)$?

1. The integration ranges for the $S(Q, \omega)$ and $X''(Q, \omega)$ are in the manuscript, but they are not so easy to find. The energy integration of ± 0.5 meV is only mentioned (I think) in the supplemental material. The L integration $[-1, 1]$ is (I believe) only mentioned in the Figure caption. Both of these can be brought out. The authors show the in-plane integration area in Fig. 1 D, but should also mention that they are integration along 110 (not 100 or 010).
2. There is no mention of potential contribution of low energy phonons in this data set. There should be some discussion as it is known that there definitely are phonons in the $14-19$ meV range at these wavevectors in LSCO, and some of these have L-dispersion which could be captured in the authors L-integration – has this been investigated? Phonons, like spin fluctuations and distinct from background, also respond to a Bose factor. So, if present, these would have to be separated from the background.
3. Was an empty can, background data set measured? How well is the “real” background understood? In the supplemental material, the authors mention “subtraction of a flat background” – does this mean Q-independent? Q and E independent? Q, E and T independent. Some discussion of the systematics of the flat background that was subtracted should be included

I have secondary concerns regarding the nature of the YSYK model and its applicability in describing inelastic neutron scattering from LSCO.

What is the physical basis of the association of the YSYK model with LSCO (at any doping)? I believe the SYK model is better known than the YSYK model – are these the same, one an extension of the other for example? My very naive understanding of the SYK model is that it is related to disordered degrees of freedom with random interactions of infinite range. Perhaps this description isn't accurate. Nonetheless I think even this reinforces my point that my read of the manuscript provided little insight into what the YSYK model was modeling, and why it can be expected to be relevant to LSCO. Some of the readership of this paper will be made up by experimentalists and others not familiar with the model. A fair amount of technical detail is provided in the supplemental material, but I think the paper would benefit from a short, physical response to this question (in the main manuscript).

Reviewer #2

(Remarks to the Author)

The authors present high-resolution inelastic neutron scattering data on overdoped cuprate LSCO which shows strange-metal behavior above its superconducting critical temperature. The data for dynamical susceptibility are shown to obey a scaling form expected of critical spin fluctuations and are consistent with a theoretical model based on Yukawa-Sachdev-Ye-Kitaev Hamiltonian in the extended Griffiths phase.

Overall these are novel and interesting results that advance our knowledge of the cuprates and more broadly a class of quantum materials that exhibit strange-metal characteristics, such as Planckian dissipation. I am in favor of acceptance once the authors had a chance to address the following minor issues:

1. In Fig. 3(a) symbols in the legend do not match symbols in the figure -- this is confusing. In addition, some symbols are open and some are solid. The meaning of this is not explained.
2. The text contains strange and sometimes ungrammatical phrasing in a number of places. An example from page 5: "along this the magnetic response reported here", or the first sentence in caption Fig. 4. Senior authors should proofread the manuscript and fix these problems.

Version 1:

Reviewer comments:

Reviewer #1

(Remarks to the Author)

My original review was very positive overall. I had several requests for clarification of the data reduction and analysis, which the authors have provided. The second referee also had a positive overall review, and minor comments which the authors have addressed. As such I now recommend acceptance in Nature Communications. This is a high quality paper.

Reviewer #2

(Remarks to the Author)

The authors have addressed the comments from the first round in a satisfactory manner. I am happy to recommend acceptance of the manuscript in the present form.

Critical spin fluctuations across the superconducting dome in $\text{La}_{2-x}\text{Sr}_x\text{CuO}_4$: response to reviewers

We thank the reviewers for their insightful comments which improve the manuscript and make it manuscript more accessible. In view of the comments we have re-organised the manuscript to include a Methods section with an explicit section on phonons which mentions the ranges of integration and other details. The previous version of the manuscript was not clear on the theory comparison. We now make it clear that the comparison is with a Hertz-Millis theory with random disorder in the tuning parameter which captures features of the YSYK model in 2D.

We respond to the points raised in turn. The Reviewer's comments are copied, followed by our response and changes made to the manuscript. The new version of the manuscript shows changes in blue.

Reviewer #1

This manuscript describes time-of-flight inelastic neutron scattering on an overdoped single crystal of LSCO $x=0.22$ and its analysis in terms of ω/T scaling and comparison to numerical calculations Yukawa-Sachdev-Ye-Kikavev model. The authors argue that this gives a picture consistent with a quantum phase transition generated by low energy, critical spin fluctuations in the presence of disorder. The argument is also made that a similar picture will generate Planckian transport with resistivity T over an extended regime of the overdoped 214 phase diagram – as have been observed. I found the experiment portion of the manuscript relatively straightforward to follow and well written. I also felt that some of what I would view as important experimental detail are relegated to the supplemental information or otherwise not easily accessible. I think that the experimental quality of the data is high, and the paper is addressing a topical and interesting subject in contemporary quantum materials physics. Provided the following points can be addressed appropriately, I feel that a suitably modified version of the manuscript is likely appropriate for Nature Communications.

It is clear that with the isolation of $X''(Q, \omega)$ shown in Fig. 1 C-F, Fig. 2 B and Fig. 3 A a very satisfactory ω/T scaling can be achieved with this data set, as shown in Fig. 3 B for LSCO $x=0.22$. A similar analysis is conveniently reproduced here in Fig. 3 C for LSCO $x=0.14$ using earlier work. From that point on the comparison to relevant theory and the idea of a broad Quantum Griffiths Phase is described and developed.

1.1 *My primary concern/criticism with this manuscript is the first step of isolating $X''(Q, \omega)$ from the measured inelastic intensity. Everything else in the manuscript depends critically on this first step being done correctly so that the magnetic scattering alone is isolated (from background, phonons) correctly. So I would like more detail included on this. This main concern really boils down to how confident are the authors that they have isolated the magnetic scattering alone in their determination of $X''(Q, \omega)$?*

Changes We have re-organised the manuscript for Nature Communications. A methods section at the end of the paper is made of material that was in the Supplementary and additional new materials to address this point. Specifically a new section on phonons, additional remarks about the background, a description of how the signal is isolated and a new figure (Supplementary Fig. 12) showing what the phonon scattering when measured for large L where it is much stronger.

1.2 *The integration ranges for the $S(Q, \omega)$ and $X''(Q, \omega)$ are in the manuscript, but they are not so easy to find. The energy integration of ± 0.5 meV is only mentioned (I think) in the supplemental material. The L integration $[-1, 1]$ is (I believe) only mentioned in the Figure caption. Both of these can be brought out. The authors show the in-plane integration area in Fig. 1 D, but should also mention that they are integration along 110 (not 100 or 010).*

Changes The integration ranges are now mentioned in the Methods section. The values the referee quotes are correct. We have mentioned explicitly that the in-plane integration is along (110) in the caption of Fig. 1 and in the Methods section.

1.3 *There is no mention of potential contribution of low energy phonons in this data set. There should be some discussion as it is known that there definitely are phonons in the 14-19 meV range at these wavevectors in LSCO, and some of these have L -dispersion which could be captured in the authors L -integration – has this been investigated? Phonons, like spin fluctuations and distinct from background, also respond to a Bose factor. So, if present, these would have to be separated from the background.*

Response There are indeed phonons in the wavevector and energy range of the experiment. We have concentrated on the energy range 0–10 meV to reduce problems with phonons such as those mentioned by the referee. The one-phonon cross-section goes as Q^2 so we have used the lowest possible L values, $|L| < 1$ to reduce phonons. We have investigated phonons at larger L in an equivalent Brillouin zone and these are shown in Supplementary Fig. 12. The form of the scattering is very different from the magnetic scattering, so the magnetic scattering and the phonons can be separated. Although phonons may show up in our scans at some energies the fit lines 'filter' them out to a large extent because of the form of the fit function. For example,

Supplementary Fig. 12b shows there is a phonon $\hbar\omega = 7$ meV and $\xi = 0$. The effect of this phonon on the analysis is best seen at $T=300$ K in Supplementary Fig. 10B ($T=300$ K, 7 meV) where the data points lie above the fit in the centre of the scan.

Changes We have included new Supplementary Fig. 12 and a section on phonons (discussing of the points raised above) in the Methods section.

1.4 *Was an empty can, background data set measured? How well is the “real” background understood? In the supplemental material, the authors mention “subtraction of a flat background” – does this mean Q-independent? Q and E independent? Q, E and T independent. Some discussion of the systematics of the flat background that was subtracted should be included.*

Response We did not do an empty can because the instrument has very low background and most of the background comes from the fairly large sample used. We believe the most of the background is incoherent phonon scattering which should increase with energy (as the phonon density states) and temperature. One can see that these trends are present in Supplementary Fig. 10. The flat background can be seen for each energy and each temperature. This was only subtracted to make the 2-D colour plots in Fig. 1C-F, Fig. 2B and Supplementary Fig. 11. When the 1-D cuts are fitted to produce the scaling plots, the background is fitted for each cut. See e.g. Supplementary Fig. 10 and the resulting plots in Fig. 3. So flat background is a Q-independent (i.e. energy a temperature dependent).

Changes In the new Method section we have rewritten the text to make clearer the procedure we used. In line with the paragraph above.

1.5 *I have secondary concerns regarding the nature of the YSYK model and its applicability in describing inelastic neutron scattering from LSCO.*

What is the physical basis of the association of the YSYK model with LSCO (at any doping)? I believe the SYK model is better known than the YSYK model – are these the same, one an extension of the other for example? My very naive understanding of the SYK model is that it is related to disordered degrees of freedom with random interactions of infinite range. Perhaps this description isn’t accurate. Nonetheless I think even this reinforces my point that my read of the manuscript provided little insight into what the YSYK model was modeling, and why it can be expected to be relevant to LSCO. Some of the readership of this paper will be made up by experimentalists and others not familiar with the model. A fair amount of technical detail is provided in the supplemental material, but I think the paper would benefit from a short, physical response to this question (in the main manuscript).

Changes The underlying model we use to describe the data is a numerical treatment of the familiar Hertz-Millis theory of spin density wave criticality in a metal in the presence of disorder. The microscopic connection of this model to the cuprates is clear, given the observation of clear peaks at the spin density wave wavevectors in our data. The particular numerical treatment we apply to this model is inspired and justified by mappings to the YSYK model which have been discussed in a number of other papers. As there is no direct use of the YSYK model in the present paper, we do not present it explicitly here. To clarify this point in the main manuscript, we have:

(i) Stated that we compare with the disordered spin density wave model not the YSYK model in the abstract and throughout the manuscript.

(ii) Expanded the discussion in the third paragraph of the “Discussion” section so the reader see which model we compare with.

Reviewer #2

The authors present high-resolution inelastic neutron scattering data on overdoped cuprate LSCO which shows strange-metal behavior above its superconducting critical temperature. The data for dynamical susceptibility are shown to obey a scaling form expected of critical spin fluctuations and are consistent with a theoretical model based on Yukawa-Sachdev-Ye-Kitaev Hamiltonian in the extended Griffiths phase.

Overall these are novel and interesting results that advance our knowledge of the cuprates and more broadly a class of quantum materials that exhibit strange-metal characteristics, such as Planckian dissipation. I am in favor of acceptance once the authors had a chance to address the following minor issues:

2.1 *In Fig. 3(a) symbols in the legend do not match symbols in the figure – this is confusing. In addition, some symbols are open and some are solid. The meaning of this is not explained.*

Changes The legend in Fig. 3(a) has been corrected to show the correct symbols. The open and closed symbols correspond to separate experiments. This is now stated in the Fig. 3 caption.

2.2 *The text contains strange and sometimes ungrammatical phrasing in a number of places. An example from page 5: “along this the magnetic response reported here”, or the first sentence in caption Fig. 4. Senior authors should proofread the manuscript and fix these problems.*

Changes Ungrammatical phrasing has been corrected in the places noted and some other places.